# Characterization and Comparative Genomics Analysis of a New Bacteriophage BUCT610 against *Klebsiella pneumoniae* and Efficacy Assessment in *Galleria mellonella* Larvae

**DOI:** 10.3390/ijms23148040

**Published:** 2022-07-21

**Authors:** Mingfang Pu, Pengjun Han, Guangye Zhang, Yucong Liu, Yahao Li, Fei Li, Mengzhe Li, Xiaoping An, Lihua Song, Yiming Chen, Huahao Fan, Yigang Tong

**Affiliations:** 1College of Life Science and Technology, Beijing University of Chemical Technology, Beijing 100029, China; 2020210710@mail.buct.edu.cn (M.P.); lyhpj88@163.com (P.H.); 4k5c111@163.com (G.Z.); TKz916545624@outlook.com (Y.L.); lf-314@163.com (F.L.); futurelmz123@163.com (M.L.); ananxiaoping@sina.com (X.A.); songlihua@mail.buct.edu.cn (L.S.); ymchen@mail.buct.edu.cn (Y.C.); 2Beijing Advanced Innovation Center for Soft Matter Science and Engineering (BAIC-SM), Beijing University of Chemical Technology, Beijing 100029, China; liyahao0036@163.com

**Keywords:** bacteriophage (phage) therapy, phage BUCT610, MDR-KP, *Galleria mellonella*

## Abstract

The spread of multidrug-resistant *Klebsiella pneumoniae* (MDR-KP) has become an emerging threat as a result of the overuse of antibiotics. Bacteriophage (phage) therapy is considered to be a promising alternative treatment for MDR-KP infection compared with antibiotic therapy. In this research, a lytic phage BUCT610 was isolated from hospital sewage. The assembled genome of BUCT610 was 46,774 bp in length, with a GC content of 48%. A total of 83 open reading frames (ORFs) and no virulence or antimicrobial resistance genes were annotated in the BUCT610 genome. Comparative genomics and phylogenetic analyses showed that BUCT610 was most closely linked with the *Vibrio* phage pYD38-A and shared 69% homology. In addition, bacteriophage BUCT610 exhibited excellent thermal stability (4–75 °C) and broad pH tolerance (pH 3–12) in the stability test. In vivo investigation results showed that BUCT610 significantly increased the survival rate of *Klebsiella pneumonia*-infected *Galleria mellonella* larvae from 13.33% to 83.33% within 72 h. In conclusion, these findings indicate that phage BUCT610 holds great promise as an alternative agent with excellent stability for the treatment of MDR-KP infection.

## 1. Introduction

*Klebsiella pneumoniae* (*K. pneumoniae*) is a Gram-negative opportunistic pathogen belonging to Enterobacteriaceae and is widely distributed in the environment, specifically in soil, surface water, domestic water, industrial wastewater, and vegetation [1]. As the one of the most common pathogens of hospital-acquired infection and community-acquired infection in the world, *K. pneumoniae* often causes pneumonia, bloodstream infection, abdominal infection, urinary system infection, and surgical wound infection [2,3,4]. Antibiotics such as ciprofloxacin, amikacin, ertapenem, imipenem, meropenem, and tigecycline are commonly used to treat infections caused by *K. pneumoniae* in clinical practice [5,6,7]. However, the spread of MDR-KP has become an emerging threat as a result of the overuse of antibiotics [8]. Bacteriophages (phages) have been known for their exclusive and strong bactericidal characteristics. Therefore, phages are also considered to be promising candidates that could replace or supplement traditional antibiotics treatment and have applications in treating infections caused by multi-resistant bacteria [9,10]. Indeed, the use of phages as antimicrobial drugs to treat severe infections caused by drug-resistant bacteria has been reported in several works. For example, in 2019, Wu Nannan et al. reported the combination of a phage and antibiotics against *K. pneumoniae* to cure a recurrent urinary tract infection caused by extensively drug-resistant *K. pneumoniae* [11]. A 62-year-old patient with a prosthetic joint infection caused by *K. pneumoniae* KpJH46 was successfully treated with phage KpJH46Φ2 combined with antibiotics at the Mayo Clinic Infectious Disease Unit in Rochester, MN, USA [12]. In addition, a 30-year-old bombing victim with a fracture-related pandrug-resistant *K. pneumoniae* infection after long-term (>700 days) antibiotic therapy was treated with a pre-adapted bacteriophage along with meropenem and colistin, followed by ceftazidime/avibactam. After three months of this combined phage–antibiotic treatment, this patient’s overall condition improved and the wound began to heal with no signs of bacterial infection [13]. These above clinical experiences with phage therapy suggest that phages combined with antibiotics may be a better treatment option for multidrug-resistant bacterial infections. In this study, the phage BUCT610 was isolated, which was capable of lysing highly virulent *K. pneumoniae* K1119. The comparative genomics of BUCT610 was analyzed using bioinformatic tools. The physiological characteristics of phage BUCT610 were investigated and the potential of BUCT610 against K1119 was evaluated in vitro and in the *Galleria mellonella* (*G. mellonella*) larvae model.

## 2. Results

### 2.1. Morphological Characteristics and Lytic Rang of Phage BUCT610

BUCT610 formed transparent, clear, regular plaques on the lawn of *K. pneumoniae* K1119 after overnight cultivation at 37 °C (Figure 1A). Transmission electron microscopy results showed that the icosahedral head diameter of BUCT610 is about 51.05 ± 2.58 nm and the tail diameter is about 145.22 ± 4.95 nm (Figure 1B). Based on the microscopic morphology of the phage virion and the classification standard of ICTV, phage BUCT610 belongs to the *Siphoviridae* family. Twenty strains of *K. pneumoniae* were collected from different hospitals, only one of which was lysed by BUCT610.

### 2.2. Multilocus Sequence Typing and Capsule Type Identification of K. pneumoniae

Seven alleles (*rpoB*, *gapA*, *mdh*, *pgi*, *phoE*, *infB*, *tonB*) were arranged (in that order) and analyzed using the multilocus sequence typing scheme that was developed for *K. pneumoniae*. According to the analysis of MLST Database, the multi-locus sequence type of host bacteria K1119 belongs to sequence type 893 (ST893), which belongs to the group of highly virulent *K. pneumoniae* (hvKp) and has been reported to spread in hospital settings centered on the Intensive Care Unit (ICU) [14,15]. No matching capsule type of K1119 was identified in the MLST database based on *wzi* allele sequencing. The multi-locus sequence type and capsule type of other *K. pneumoniae* are shown in Table 1.

### 2.3. Lifestyle of BUCT610

The qPCR results showed that none of the three genes of phage BUCT610 were detected in the genome of R-K1119, indicating that the genome of phage BUCT610 cannot be integrated into the host K1119 genome. The genome of a temperate phage can be integrated into the host genome [16]. Therefore, phage BUCT610 is a lytic phage rather than a temperate phage. The results of qPCR are shown in the Appendix A.

### 2.4. Physiological Characterization of Phage BUCT610

The virion production of phages BUCT610 is significantly higher than other groups when the MOI is 0.01, indicating that 0.01 is the most suitable MOI for the growth of BUCT610 (Figure 2A). The latent period and lytic period of phage BUCT610 were approximately 30 min and 60 min, respectively (Figure 2B). Thermal stability tests showed that the titer of phage BUCT610 was essentially constant from 4 °C to 65 °C during the same incubation time, while the titer of phage decreased significantly at 75 °C (Figure 2C). The titer of phage BUCT610 was essentially constant in different pH (3–12) environments during the same incubation time, expect at pH = 2, where the phage titer decreased sharply (Figure 2D).

### 2.5. Annotations and Characteristics of the Complete Genome of BUCT610

The full length of linear double-stranded DNA of phage BUCT610 is 46,774 bp, with a GC content of 48%, and its accession number is MZ318367 in the NCBI database. According to RAST and BLASTp analysis results, a total of 83 open reading frames (ORFs) were predicted in the BUCT610 complete genome; 13 of them encode proteins with functions associated with phage lysis, packaging, structure, and replication (Figure 3, Table 2). The other ORFs were annotated as encoding hypothetical proteins. In addition, no virulence, antimicrobial resistance, or tRNA genes are annotated in the BUCT610 genome. The complete genome sequence alignment of phage BUCT610 with the complete genome sequences of other phages in the NCBI data showed that BUCT610 shared the highest homology (cover: 69%, identity: 92.9%) with the *Vibrio* phage pYD38-A (GenBank: JF974312.1) (Figure 4A), followed by the *Aeromonas* phage pIS4-A (GenBank: NC_042037.1) (Figure 4B), the *Klebsiella* virus KpV2811 (GenBank: NC_054653.1) (Figure 4C), and Flyfo siphovirus Tbat1_6 (GenBank: OL617041.1) (Figure 4D), which shared 69%, 65%, and 65% cover and 93.02%, 91.82%, and 91.76% identity, respectively. The average nucleotide identity (ANI) value reflects the relationship between the two species. The average nucleotide identity (ANI) values were calculated using the reciprocal best hits (two-way ANI) between two genomic datasets. The ANI values between the phage BUCT610 and the *Vibrio* phage pYD38-A (GenBank: JF974312.1), *Aeromonas* phage pIS4-A (GenBank: NC_042037.1), and the *Klebsiella* virus KpV2811 were 92.29%, 92.38%, and 90.81%, respectively. This result indicates that phage BUCT610 is a member of a new species according to the introduction of phage classification [17]. In addition, the complete genome phylogenetic tree of BUCT610 and twenty-five other phages showed that BUCT610 shared a close evolutionary relationship with Flyfo *siphovirus* Tbat1_6 (GenBank: OL617041.1) and the *Klebsiella* phage vB_KpnS_MK54 (GenBank: NC_054652.1) (Figure 5A).

### 2.6. Replication-, Transcription-, and Structure-Related Genes of Phage BUCT610

The annotations of the BUCT610 genome sequence showed that a total of 13 ORFs encoded proteins functionally related to the replication, translation, and structure of the phage BUCT610. ORF64, ORF65, and ORF68 of the BUT610 were predicted as recombinases, exonucleases, and DNA helicases, respectively. Recombinase is adjacent to exonuclease and DNA helicases, which are three enzymes associated with the replication of phage genomes [18]. The recombinant enzyme was responsible for recognizing and cutting specific recombination sites and connecting the two molecules involved in the recombination. Recombinase has a close connection with the mechanism of phage DNA replication [19]. The homologous recombination systems of linear double-stranded (ds) DNA bacteriophages were required for the generation of genetic diversity, the repair of dsDNA breaks, and the formation of concatemeric chromosomes [20]. DNA exonuclease is a multifunctional hydrolase. Such as Exonuclease 1 (EXO1) plays a role in DNA replication, DNA mismatch repair, and DNA double-strand break repair (DSBR) and has been associated with increased susceptibility to certain cancers [21]. ORF16 and ORF59 are predicted to be lysozyme and C40 family peptidase, and these two proteins are classified as lytic proteins based on their functions. Lysozyme is an alkaline enzyme that can hydrolyze mucopolysaccharides in bacteria, mainly destroys the β-1,4 glycoside bond between N-acetylmuramic acid and N-acetylglucosamine in the cell wall, and decomposes the insoluble mucopolysaccharide into soluble glycopeptide, resulting in cell wall rupture and bacterial dissolution. Peptidase is a type of enzyme that hydrolyzes peptide chains; it mainly exists in microbial cells and can be divided into two categories: endopeptidase and exopeptidase.

ORF30 of phage BUCT610 is predicted to encode a coat protein. As a carrier tool, coat protein is widely used by phage display technology [22,23]. In addition, portal protein coded by ORF10 and tail length tape measure protein coded by ORF56 were also found in the BUCT610 genome annotation. Whether the virus can successfully infect the cell depends on whether the virus genome was successfully transmitted to the host cell. The portal protein is one of the key proteins involved in virus replication. During the packaging of the phage P22 genome, portal proteins are involved in the maturation of virions through conformational changes. The tail length tape measure protein is so named since the length of the corresponding gene is proportional to the length of the bacteriophage tail [24,25].

ORF83 of BUCT610 was annotated as terminase large subunit whose main function is to package bacteriophage genomic DNA into empty capsid proteins and confer most of the enzymatic activities required in the process of DNA packaging [26], including ATP enzyme, endonuclease, and DNA helicase activity [27,28,29]. The terminase subunit was widely found in double-stranded DNA phages; as one of the conserved proteins of phages, it has often been used to investigate phage evolutionary relationships. A phylogenetic tree constructed based on the terminase sequence of the phage BUCT610 and thirty-two other phage terminase sequences in the NCBI data showed that the BUCT610 terminase protein is closely related to the terminase proteins of the *Klebsiella* phage ZCKP8 and *Klebsiella* phage vB_KpnS_MK54 (Figure 5B).

### 2.7. Assessment the Efficacy of BUCT610 against K1119 In Vitro and In Vivo

Phage BUCT610 with titers of 5 × 10^9^ PFU/mL and 5 × 10^8^ PFU/mL significantly inhibited the growth of K1119 compared to the positive control (Figure 6A). Survival rates of *K. pneumoniae* K1119-infected *G. mellonella* larvae were 73.33%, 80%, 83.33%, and 50% within 72 h when the titers of BUCT610 were 3 × 10^7^ (PFU/each, MOI = 100), 3 × 10^6^ (PFU/each, MOI = 10), 3 × 10^5^ (PFU/each, MOI = 1), and 3 × 10^4^ (PFU/each, MOI = 0.1), respectively (Figure 6B,C). The survival rate of *G. mellonella* larvae in the positive control group was only 13.33%, while all the larvae in the negative control group were alive.

## 3. Discussion

We obtained a *K. pneumoniae* from hospital ward environments and named it K1119. K1119 belongs to *K. pneumoniae* ST893 according to the MLST of *K. pneumoniae*. Notably, no capsule type matched with K1119 in the MLST database based on the *wzi* allele sequencing identification, indicating that K1119 is a *K. pneumoniae* ST893 with a unique capsule type. Therefore, it is beneficial to find alternative agents that can replace antibiotics to deter the spread of K1119. Here, we isolated a lytic phage BUCT610, which can form clear, regular plaques on the bacterial lawns containing K1119. The phage BUCT610 is not a temperate phage, its genome cannot integrate into the K1119 genome, although it can only lyse *K. pneumoniae* K1119. According to reports related to the recognition mechanism of phages and hosts, the lipopolysaccharides (LPS) and capsular polysaccharide (CPS) of bacteria play an important role in phage adsorption, which is an important factor in determining the host range of phages [30,31]. Therefore, we speculate that the specific capsule type of K1119 is one of the main reasons for the phage BUCT610 to specifically lyse *K. pneumoniae* K1119. Although phage BUCT610 only lyses K1119 at present, there is no barrier against the combination of phage BUCT610 and antibiotics or BUCT610 and other phages as a cocktail against MDR-KP [32,33].

The genome characteristics and safety of phages can be analyzed through bioinformatic tools. The BUCT610 genome length is 46,774 bp, with 48% GC. A total of 13 functional proteins are encoded throughout the genome, and these functional proteins are involved in the replication of the phage BUCT610 genome as well as in the stabilization of the structure. However, a large number of hypothetical proteins were annotated in the complete genome of BUCT610 and further studies are needed. It is notable that no virulence or antimicrobial resistance genes are annotated in the genome of phage BUCT610, which indicates that phage BUCT610 is theoretically safe as an antimicrobial agent. A phylogenetic tree was constructed based on the complete genome sequence of phage BUCT610 and other phage complete genome sequences in the NCBI database, showing that BUCT610 has a close evolutionary relationship not only with the *Klebsiella* phage vB_KpnS_MK54 (GeneBank: MW119258.1), but also with Flyfo siphovirus Tbat1_6 (GeneBank: OL617041.1). In response to host selection pressure, phage BUCT610 and Flyfo siphovirus Tbat1_6 may undergo multiple gene exchange events which drive their diversity [34,35].

The activity and stability of phages are closely related to the preparation and application of phages. BUCT610 has a broad range of pH tolerance and good thermal stability. The titer of phage BUCT610 remained essentially the same as the titer at pH = 7 in both extreme acidic (pH = 3) and alkaline (pH = 12) environments compared to *K. pneumoniae* phage VTCCBPA43, which was completely inactive at pH = 3 [36]. Furthermore, phage BUCT610 has excellent thermal stability. The titer of phage BUCT610 remained essentially unchanged at 4 °C compared to 65 °C, while most phages can only maintain good stability from 4 °C to 60 °C [37]. The favorable stability of phage BUCT610 facilitates the storage of the phage and gives it potential as an antimicrobial agent [38].

Phage therapy is attracting attention as an alternative therapy compared to traditional antibiotic therapy. BUCT610 has outstanding stability and has been demonstrated as an efficient anti-K1119 agent in vitro. *G. mellonella* larvae have been widely used as a model because they are cheap and pose few ethical problems compared with other models. We further evaluated the efficacy of phage BUCT610 in treating the *G. mellonella* larvae infected with *K. pneumoniae* K1119. A titer of 3 × 10^6^ (PFU/each, MOI = 10) of BUCT610 increased the survival rate of *G. mellonella* larvae from 13.33% to 83.33% within 72 h compared to the positive control group. BUCT610 with a titer of 3 × 10^5^ (PFU/each, MOI = 1) showed the best treatment efficacy for *G. mellonella* larvae within 14–44 h, but the treatment efficacy within 44–72 h was slightly lower than that when MOI = 10. Interestingly, the survival rate of *G. mellonella* larvae was only 73.33% for the highest titer (3 × 10^7^ PFU/each, MOI = 100) of phage BUCT610-treated larvae infected with K1119, which was lower than the survival rate of *G. mellonella* larvae when MOI = 10 and MOI = 1. The survival rate of the *G. mellonella* larvae does not increase with higher titers of phage BUCT610. We speculate that the high-titer phage BUCT610 removed K1119 from *G. mellonella* larvae, K1119 may have produced a large amount of endotoxin, and these endotoxins caused the death of some larvae [39,40,41]. This result strongly suggests the flexibility of phage titers during phage treatment, especially for infections caused by Gram-negative bacteria.

In conclusion, we characterized BUCT610 as a lytic phage against *K. pneumoniae* ST893. The bioinformatic analysis results and evaluation of therapeutic efficacy in *G. mellonella* larvae show that phage BUCT610 has the potential to effectively constrain the spread of novel *K. pneumoniae*.

## 4. Materials and Methods

### 4.1. Bacterial Strain Cultures

*K. pneumoniae* K1119 was obtained from environmental samples of the 307 hospital, Beijing. The strains used in this study were stored at −80 °C in 50% glycerol (*v*/*v*) and cultured in the lysogeny broth (LB) at 37 °C [42]. The viable bacterial count was determined by finite gradient dilution method using LB agar (1.5% *w*/*v*) plates.

### 4.2. Bacteriophage Isolation

Sewage samples from hospitals were enriched with *K. pneumoniae*. Briefly, centrifuged sewage samples were filtered with a sterile 0.22 μm filter and co-cultured with *K. pneumoniae* at 37 °C for 6 h. The supernatant containing the phages was collected, filtered using a sterile 0.22 μm filter, and finally spotted on a double-layer agar plate of *K. pneumoniae* strain K1119. The phage plaques were amplified in plates as described previously, with slight modifications [43]. Single plaque was picked up and co-cultured with *K. pneumoniae* K1119 in LB medium for 8 h. After centrifugation, the supernatant was filtered, and the filtrate was diluted properly and separated by a double-layer plate. The above steps were repeated three times, and the isolated bacteriophages were amplified using LB liquid medium at 37 °C and 220 rpm and stored at 4 °C.

### 4.3. Purification of Phage BUCT610

The isolated phage BUCT610 and host K1119 were co-cultured at 37 °C for 8 h and centrifuged at 4 °C for 12,000× *g* for 15 min to remove cells and cell debris. Crude phage was obtained after the supernatant was filtered by the sterile 0.22 μm filter. The crude phage was further purified and concentrated with 30% sterile sucrose solution by centrifugation at 30,000 rpm and 4 °C for 2 h [44,45]. The supernatant after centrifugation was discarded and the bottom phage precipitate was resuspended with phosphate buffer saline (PBS) to obtain pure phage BUCT610, which was then stored at 4 °C.

### 4.4. Electron Microscopy

To visualize the phages, a carbon-coated copper grid was incubated with 30 µL phage particle for 10 min and stained with 2% uranyl acetate for 90 s [46]. The morphology of the phages was examined with a transmission electron microscope (JEM-1200EX, JEOL, Tokyo, Japan) at 80 kV.

### 4.5. Identified Lytic Range of Phage BUCT610

Twenty strains of *K. pneumoniae* were collected from different hospital ward environments and were cultured to the exponential phase. The lytic range of phage BUCT610 was determined by the double layer plate method and spotting method [47].

### 4.6. Multilocus Sequence Typing (MLST) and Capsule Type of K. pneumoniae

We aimed to identify the subtypes of the twenty strains of *K. pneumoniae* which were used for lytic-range testing. The MLST and capsule type of *K. pneumoniae* were identified. The determination methods of MLST and capsule type for *K. pneumoniae* were similar to that described by Yannan Liu et al., with slight modifications [48]. In brief, seven housekeeping genes (*rpoB*, *gapA*, *mdh*, *pgi*, *phoE*, *infB*, and *tonB*) and *wzi* allele gene of the twenty *K. pneumoniae* were subjected to PCR amplification (PCR procedure for MLST identification: step 1: 94 °C for 2 min; step 2: 94 °C for 20 s; step 3: 50 °C for 30 s; step 4: 72 °C for 30 s; step 5: 72 °C for 5 min; step 4 to step 2: 35 cycles; PCR procedure for capsule type identification: step 1: 94 °C for 2 min; step 2: 94 °C for 30 s; step 3: 55 °C for 40 s; step 4: 72 °C for 30 s; step 5: 72 °C for 5 min; step 4 to step 2: 35 cycles). The amplified products were sent to Beijing Ruibo Xingke Biotechnology Co., Ltd. for bidirectional sequencing. The sequencing results were analyzed by the MLST database (https://bigsdb.pasteur.fr/cgi-bin/bigsdb/bigsdb.pl?db=pubmlst_klebsiella_seqdef&l=1 accessed on 22 May 2022) for analysis. Primer sequences are shown in Table 3.

### 4.7. Lifestyle Identification of BUCT610

We aimed to investigate whether the phage BUCT610 genome could be integrated into the host K1119 genome. Four strains of *K. pneumoniae* resistant to phage BUCT610 (R-K1119) were obtained from the co-cultures of phage BUCT610 and host K1119. The R-K1119 were validated and further purified by the spot plate method and plate streaking. Three ORFs of the phage BUCT610 were detected by real time PCR (qPCR) using the R-K1119 as the template. The positive and negative groups used phage BUCT610 and K1119 as templates, respectively. The three tested ORFs of BUCT610 were ORF64 (recombinase), ORF70 (conserved hypothetical protein), and ORF83 (large terminase subunit), respectively. The information for primers is shown in Table 3.

### 4.8. Optimal Multiplicity of Infection and One-Step Growth Curve of BUCT610

To determine the optimal multiplicity of infection (MOI), serial dilutions of *K. pneumoniae* K1119 were added to aliquots of BUCT610 (10^6^ PFU/mL). A volume of 100 μL of the mixture with different MOI (0.001, 0.01, 0.1, 1, 10, 100) was added into 10 mL LB medium for overnight culture at 37 °C with shaking (220 rpm). Then, the mixture was centrifuged at 12,000× *g* for 2 min at room temperature to remove residual bacterial cells. The supernatant was filtered through a 0.22 µm filter. The titer of the phage was determined by the double-layer plate method, and the experiments were repeated three times. To determine the One-step growth curve of phage BUCT610, the mixture of BUCT610 and host K1119 was incubated for 10 min at room temperature with optimal MOI. The supernatant was discarded after centrifugation at 4 °C for 3 min at 12,000× *g.* The precipitate was resuspended with LB and then centrifuged for 3 min at 4 °C and 12,000× *g*, and repeat the above steps again. Mixture after resuspension was added to 25 mL of LB liquid medium and cultured at 37 °C for 150 min with shaking 200 rpm. The titer of phage BUCT610 was detected at different time points. The above experiments were performed on ice. Titer of BUCT610 at different time points were detected by double-layer plate method.

### 4.9. Stability Assessment of Phage BUCT610

The stability of phage BUCT610 was performed as described previously with slight modifications [49]. BUCT610 was incubated at different temperatures (4 °C, 37 °C, 45 °C, 55 °C, 65 °C, 75 °C) for 1 h. The titer of phage BUCT610 was detected after incubation 1 h. In order to ascertain the stability of the bacteriophage in different pH environments, the phage BUCT610 was incubated at different pH levels ranging from 2 to 12 for 1 h. As mentioned above, the phage titers were detected after incubation under the different pH buffers. The different pH buffers were adjusted by hydrochloric acid and sodium hydroxide.

### 4.10. Phage Sequencing and Bioinformatics Analysis

The genomic DNA of the bacteriophage BUCT610 was extracted by the classical proteinase K/SDS method and the concentration was measured by Quantum Bit Fluorometer (Qubit™ Fluorometer) [47,50]. According to the instructions, 150 ng of genomic DNA was invested in the NEBNext Ultra II FS DNA Library Prep Kit (NEB). The genomic DNA was fragmented, indexed by PCR, and purified using AMPure XP beads. Then, the Agilent 2100 BioAnalyzer system was used to evaluate the size distribution of the constructed library and the library was quantified by KAPA Library Quantification Kits. Finally, the assembled DNA library was sequenced by Illumina Novaseq, and 2 × 150 bp paired-end reads were generated.

Quality control software FastQC V0.11.5 was used to analyze the quality of the original sequence data, and Trimmomatic 0.36, the default parameter, was used to remove the low-quality reads and adapter regions [51,52]. High-quality reads were assembled using SPAdes v3.13.0 [53]. Potential protein-coding genes were predicted using RAST, and the protein functions were annotated using the protein basic local alignment search tool (BLASTp) against the non-redundant protein sequences (nr) database [54]. Genome functional maps were produced by a lab-built script and modified using Inkscape 0.92.3.0. The molecular weight of proteins encoded by the open reading frames (ORFs) was determined using the ExPASY ProtParam tool [55]. The recognized tRNA coding genes were searched by tRNAscan-SE v.2.0 [56,57]. Antimicrobial resistance genes and virulence factors in the phage genome were predicted using an online prediction platform ResFinder and VirulenceFinder, respectively [58,59,60]. The phylogenetic tree based on the large terminase was constructed by the neighbor joining method (NJ method) and detected by the bootstrap method with 1000 test repetitions. Average nucleotide identity (ANI) values were calculated using an ANI calculator. Complete genome sequence similarity BUCT610 and other phages was visualized by Circoletto program (Version number: 07.09.16) [49].

### 4.11. Assessment the Efficacy of BUCT610 against K. pneumoniae K1119 In Vitro

The 25 mL K1119 was cultured to OD_600_ of about 0.2 at 37 °C and 120 rpm. The 5 mL of phage BUCT610 with titers 5 × 10^9^ PFU/mL and 5 × 10^8^ PFU/mL were mixed with 25 mL K1119, respectively. The mixture continued to incubate at 37 °C and 120 rpm. The variations of OD_600_ of the mixture were detected during cultivation.

### 4.12. Therapeutic Effect of Phage BUCT610 in the G. mellonella Larvae

*G. mellonella* larvae (Huiyude Biotech Company, Tianjin, China) were selected with length 25 ± 5 mm, weight 300 ± 50 mg, strong activity, and no black patches on the body surface. Thirty larvae were used as a sample population for each group. *K. pneumoniae* K1119 were grown in LB and harvested in the exponential phase. After being washed with PBS, 6 μL K1119 (5 × 10^7^ CFU/mL) was injected into the last right proleg of larvae by a microsample syringe, and half an hour after infection, phage BUCT610 was injected into the last left proleg of larvae for treatment. The number of surviving *G. mellonella* larvae at different MOIs groups was observed and recorded every 2 h for 72 h. The positive control group was treated with PBS (6 μL/each), and only PBS or phage BUCT610 was injected as the negative group. Results were considered valid when all larvae of the PBS-injected samples survived during the experiment. The above experiment was repeated three times.

## Figures and Tables

**Figure 1 ijms-23-08040-f001:**
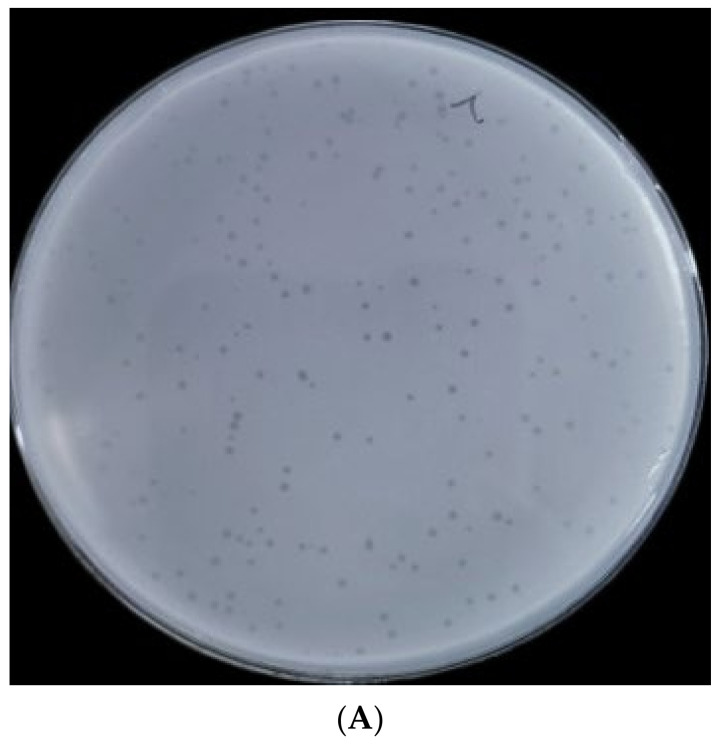
Plaque morphology of bacteriophage BUCT610. (**A**) Phage plaques formed of BUCT610 on the lawn with *K. pneumoniae* K1119. (**B**) Transmission electron micrograph image of BUCT610.

**Figure 2 ijms-23-08040-f002:**
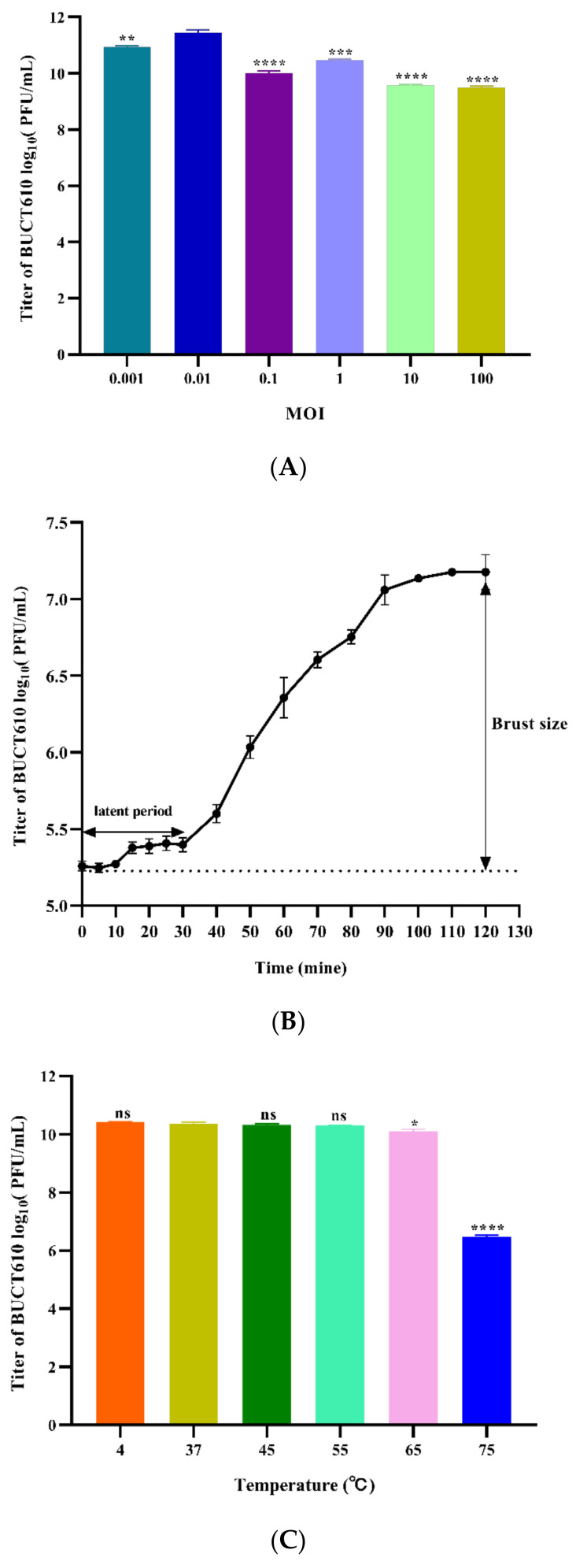
Physiological characterization of phage BUCT610. (**A**) Optimal MOI assays of bacteriophage BUCT610. (**B**) One-step growth curve of bacteriophage BUCT610. (**C**) Thermal stability of BUCT610. (**D**) pH stability of bacteriophage BUCT610. Data are shown as the mean ± SD; **** *p* < 0.0001, *** *p* < 0.001, ** *p* < 0.01, or * *p* < 0.05 indicates a significant difference between this group and the control (this experiment was repeated in biological triplicate).

**Figure 3 ijms-23-08040-f003:**
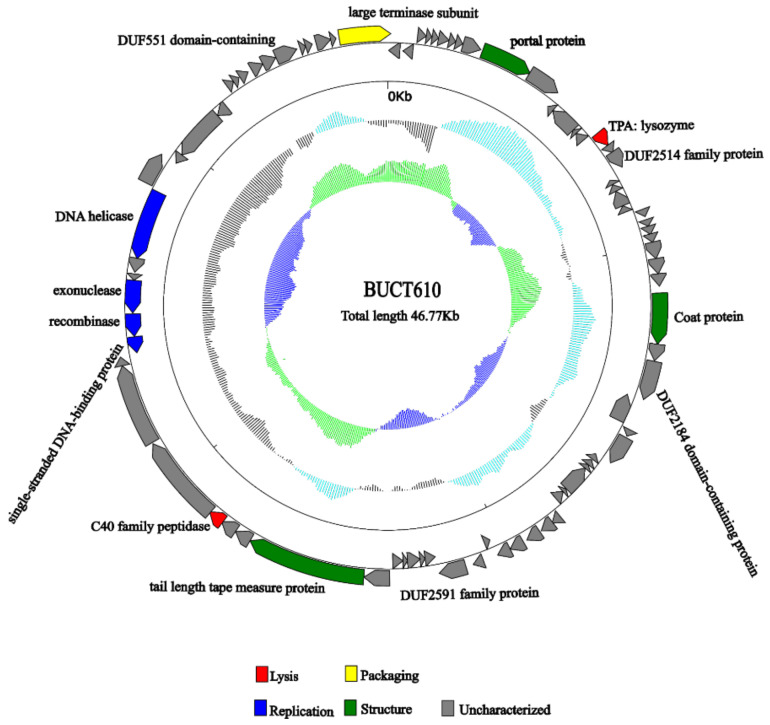
Genomic map of bacteriophage BUCT610 and its genetic characteristics. Open reading frames (ORF) are represented in different colors according to their functional categories. Red: lysis-related genes; blue: replication-related genes; green: structure-related genes; yellow: packaging-related genes; gray: uncharacterized genes.

**Figure 4 ijms-23-08040-f004:**
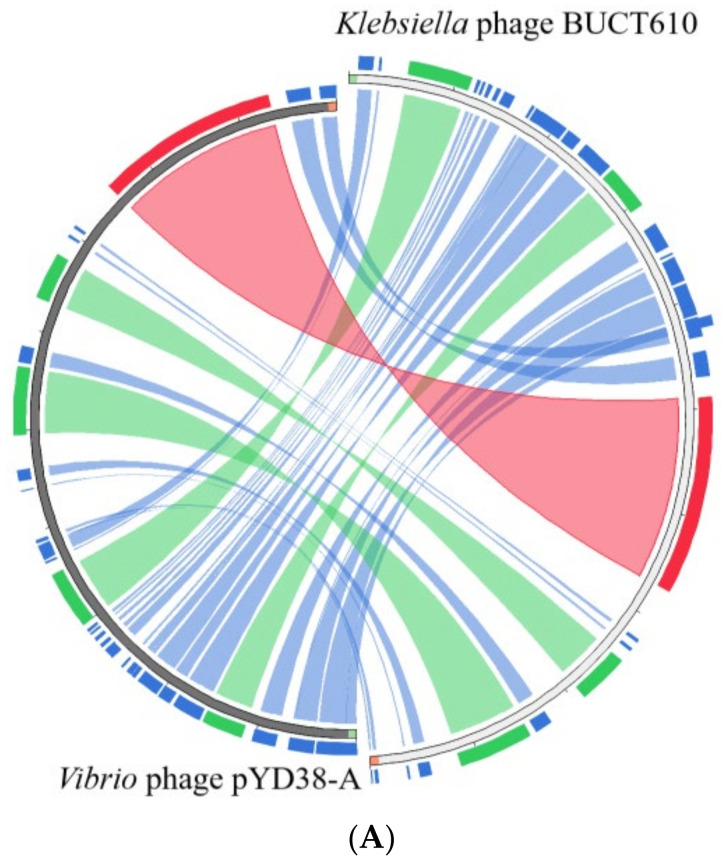
Circos plot depicting sequence similarities of *Klebsiella* phage BUCT610 with (**A**) *Vibrio* phage pYD38-A, (**B**) *Aeromonas* phage pIS4-A, (**C**) *Klebsiella* virus KpV2811, and (**D**) Flyfo siphovirus Tbat1_6. The red color signifies a high sequence similarity, followed by orange, green, and blue. Ratio coloring: blue, ≤0.25; green, ≤0.50; orange, ≤0.75; red, >0.75.

**Figure 5 ijms-23-08040-f005:**
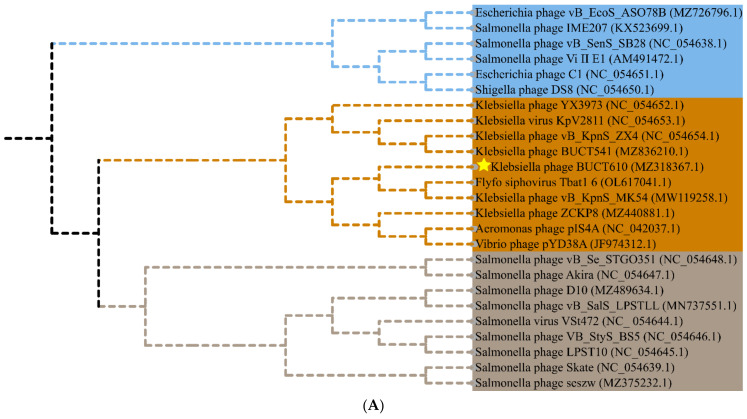
Phylogenetics tree of bacteriophage BUCT610. (**A**) Phylogenetic trees of *Klebsiella* phage BUCT610 based on the complete genome sequence generated by VICTOR. (**B**) Phylogenetic trees were constructed based on the terminase large subunit of BUCT610. The amino acid sequences of the related phages were downloaded from NCBI.

**Figure 6 ijms-23-08040-f006:**
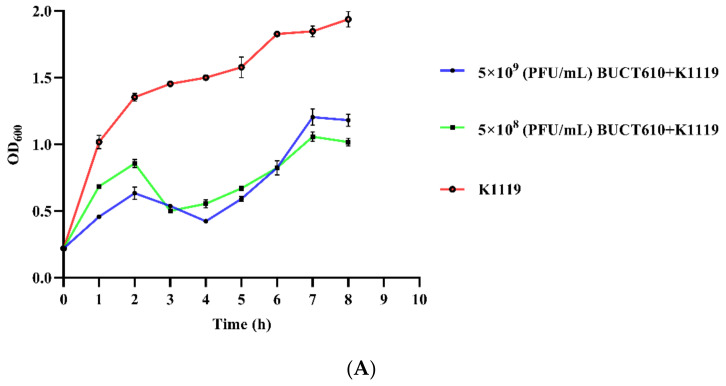
Assessment the efficacy of BUCT610 against K1119 in vitro and in vivo. (**A**) The variation of OD_600_ when different titers of BUCT610 were co-cultured with K1119. (**B**) Survival rate of *G. mellonella* larvae. Treatment of the *K. pneumoniae* K1119 (3 × 10^5^ CFU/each)-infected larvae at different titers of BUCT610 (this experiment was repeated in biological triplicate). (**C**) Different titers of phage BUCT610 therapy for *G. mellonella* larvae infected with K1119.

**Table 1 ijms-23-08040-t001:** Detailed information for bacteria of host range tested.

Species	Strain	ST Type	Capsule Type	Origin	Sensitivity
*K. pneumoniae*	K1119	ST893	No match	307 hospital	+
*K. pneumoniae*	K7	ST23	K1	Aviation General Hospital	-
*K. pneumoniae*	064	ST23	K1	Aviation General Hospital	-
*K. pneumoniae*	081	ST23	K1	Aviation General Hospital	-
*K. pneumoniae*	1241	ST11	K15	307 hospital	-
*K. pneumoniae*	1291	ST11	K27	307 hospital	-
*K. pneumoniae*	1299	ST723	N/A	307 hospital	-
*K. pneumoniae*	1300	ST11	K16	307 hospital	-
*K. pneumoniae*	1301	ST65	K2	307 hospital	-
*K. pneumoniae*	1307	ST65	K2	307 hospital	-
*K. pneumoniae*	2012	ST11	K25	Aviation General Hospital	-
*K. pneumoniae*	2013	ST15	K19	Aviation General Hospital	-
*K. pneumoniae*	2015	ST11	K47	Aviation General Hospital	-
*K. pneumoniae*	2017	ST15	K19	Aviation General Hospital	-
*K. pneumoniae*	2019	N/A	K47	Aviation General Hospital	-
*K. pneumoniae*	2021	N/A	K64	Aviation General Hospital	-
*K. pneumoniae*	2068	ST15	K19	Aviation General Hospital	-
*K. pneumoniae*	2086	ST11	K64	Aviation General Hospital	-
*K. pneumoniae*	2752	ST23	K1	Aviation General Hospital	-
*K. pneumoniae*	2755	ST23	K1	Aviation General Hospital	-

“+” susceptible; “-” resistance; “N/A” mutation.

**Table 2 ijms-23-08040-t002:** Predicted ORFs in the genome of phage BUCT610.

ORF	Strand	Start	Stop	Length (AA)	Putative Function	Best-Match BLASTp Result	Query Cover (%)	E-Values	Identity (%)	Accession	MW (kDa)
ORF1	-	352	20	110	hypothetical protein	*Klebsiella* phage KMI7	98%	7.00E-41	62.04%	QEG10317.1	13.14
ORF2	-	729	439	96	hypothetical protein	*Klebsiella* phage YX3973	100%	2.00E-62	98.96%	YP_010054377.1	10.70
ORF3	+	818	1057	79	hypothetical protein	*Klebsiella* phage ZX4	100%	3.00E-48	97.47%	YP_010054495.1	9.32
ORF4	+	1054	1206	50	hypothetical protein	*Trabulsiella odontotermitis*	96%	1.00E-06	50.00%	WP_158674504.1	5.67
ORF5	+	1215	1394	59	hypothetical protein	*Salmonella* phage SeSz-2	100%	1.00E-14	61.02%	YP_010053632.1	6.84
ORF6	+	1394	1690	98	hypothetical protein	*Pectinobacterium* phage PEAT2	95%	1.00E-37	63.83%	YP_009702196.1	11.68
ORF7	+	1687	1875	62	hypothetical protein	*Escherichia coli*	100%	4.00E-24	70.97%	MBB0589830.1	6.70
ORF8	+	1875	2075	66	hypothetical protein	*Salmonella enterica* subsp	100%	4.00E-07	34.85%	HAZ2983796.1	7.66
ORF9	+	2085	2597	170	hypothetical protein	*Salmonella enterica*	97%	6.00E-64	57.74%	EGT9551729.1	19.34
ORF10	+	2646	4094	482	portal protein	*Klebsiella* phage ZX4	100%	0	97.30%	YP_010054497.1	54.12
ORF11	+	4075	5019	314	hypothetical protein	*Klebsiella* virus KpV2811	100%	0	97.13%	YP_010054477.1	35.56
ORF12	-	5226	5032	64	hypothetical protein	*Klebsiella* phage YX3973	100%	5.00E-31	82.81%	YP_010054385.1	7.53
ORF13	-	6049	5228	273	hypothetical protein	*Klebsiella* phage YX3973	100%	5.00E-91	56.12%	YP_010054386.1	30.50
ORF14	-	6177	6046	43	hypothetical protein	*Klebsiella* phage ZX4	100%	3.00E-21	93.02%	YP_010054501.1	5.12
ORF15	-	6467	6177	96	hypothetical protein	*Klebsiella* virus KpV2811	100%	1.00E-57	94.79%	YP_010054472.1	9.95
ORF16	+	6566	7003	145	TPA: lysozyme	*Salmonella enterica* subsp	100%	3E-103	97.93%	HAZ2983869.1	15.70
ORF17	+	7000	7245	81	hypothetical protein	*K. pneumoniae*	100%	3.00E-51	100.00%	WP_142483241.1	9.06
ORF18	+	7220	7717	165	DUF2514 family protein	*Klebsiella* virus KpV2811	96%	3.00E-91	93.71%	YP_010054469.1	18.13
ORF19	-	8013	7864	49	hypothetical protein	*K. pneumoniae*	100%	5.00E-25	95.92%	WP_185939371.1	5.64
ORF20	-	8233	8021	70	hypothetical protein	*Klebsiella* phage YX3973	100%	1.00E-40	91.43%	YP_010054391.1	8.20
ORF21	-	8700	8233	155	hypothetical protein	*Vibrio* phage pYD38-A	100%	7.00E-91	86.16%	YP_008126189.1	17.50
ORF22	-	8900	8700	66	hypothetical protein	*K. pneumoniae*	100%	2.00E-40	100.00%	WP_142483236.1	8.02
ORF23	+	8997	9212	71	hypothetical protein	*Klebsiella* phage YX3973	100%	2.00E-44	100.00%	YP_010054394.1	8.26
ORF24	+	9296	9487	63	hypothetical protein	*K. pneumoniae*	100%	1.00E-35	96.83%	WP_142483234.1	7.70
ORF25	+	9484	9723	79	hypothetical protein	*Vibrio* phage pYD38-A	100%	6.00E-35	89.87%	YP_008126184.1	9.62
ORF26	+	9704	9946	80	hypothetical protein	*Salmonella* phage SeSz-2	97%	8.00E-28	66.67%	YP_010053654.1	8.97
ORF27	+	9936	10,406	156	hypothetical protein	*Klebsiella* phage YX3973	100%	8E-113	100.00%	YP_010054398.1	17.67
ORF28	+	10,394	10,819	141	hypothetical protein	*Shigella* phage Sf11 SMD-2017	55%	1.00E-39	80.77%	YP_010053088.1	16.52
ORF29	+	10,816	11,160	167	hypothetical protein	*Klebsiella* phage YX3973	68%	8.00E-74	93.86%	YP_010054400.1	13.05
ORF30	+	11,343	12,746	467	hypothetical protein	*Klebsiella* phage YX3973	100%	0	96.15%	YP_010054401.1	51.39
ORF31	+	12,746	13,210	207	hypothetical protein	*Vibrio* phage pYD38-A	74%	7E-103	97.40%	YP_008126178.1	15.64
ORF32	+	13,222	14,310	362	DUF2184 domain containing protein	*K. pneumoniae*	100%	0	99.45%	WP_142483227.1	40.05
ORF33	-	15,123	14,326	265	hypothetical protein	*Salmonella enterica *	100%	5E-116	60.97%	ECD2278621.1	31.84
ORF34	+	15,215	15,409	64	hypothetical protein	*Vibrio* phage pYD38-A	100%	6.00E-37	98.44%	YP_008126175.1	7.11
ORF35	+	15,453	16,271	272	hypothetical protein	*K. pneumoniae*	94%	1E-180	97.66%	WP_142483225.1	28.65
ORF36	-	16,483	16,301	60	hypothetical protein	*Salmonella* phage IME207	100%	2.00E-28	83.33%	YP_009322759.1	6.76
ORF37	-	16,682	16,509	57	hypothetical protein	*K. pneumoniae*	100%	4.00E-33	96.49%	WP_185939376.1	6.71
ORF38	-	16,819	16,682	45	hypothetical protein	*Bacteriophage* sp.	97%	3.00E-15	68.18%	QHJ81118.1	5.27
ORF39	-	17,634	16,819	271	hypothetical protein	*K. pneumoniae*	100%	0	98.52%	WP_142483222.1	29.94
ORF40	-	17,758	17,642	38	hypothetical protein	*Klebsiella* phage ZX4	100%	5.00E-14	92.11%	YP_010054525.1	4.42
ORF41	-	18,048	17,755	97	hypothetical protein	*Vibrio* phage pYD38-A	100%	6.00E-22	53.61%	YP_008126169.1	10.02
ORF42	+	18,279	18,566	95	hypothetical protein	*Klebsiella* phage YX3973	100%	5.00E-62	100.00%	YP_010054411.1	10.16
ORF43	+	18,544	18,936	130	hypothetical protein	*Klebsiella* phage YX3973	100%	9.00E-89	97.69%	YP_010054412.1	13.88
ORF44	+	18,974	19,399	141	hypothetical protein	*Vibrio* phage pYD38-A	80%	8.00E-79	100.00%	YP_008126166.1	15.18
ORF45	+	19,504	19,920	138	hypothetical protein	*Klebsiella* virus KpV2811	100%	5.00E-94	98.55%	YP_010054440.1	14.99
ORF46	+	19,917	20,300	127	hypothetical protein	*Vibrio* phage pYD38-A	100%	6.00E-89	100.00%	YP_008126240.1	14.47
ORF47	-	20,509	20,324	61	hypothetical protein	*Klebsiella* virus KpV2811	100%	1.00E-35	95.08%	YP_010054438.1	7.06
ORF48	+	20,713	21,015	100	hypothetical protein	*Pectobacterium polonicum*	39%	9.00E-09	82.05%	WP_137741933.1	11.10
ORF49	+	21,219	21,974	251	hypothetical protein	*K. pneumoniae*	99%	2e-178	99.60%	WP_142483279.1	26.54
ORF50	-	22,298	21,996	100	hypothetical protein	*Salmonella enterica*	100%	5.00E-55	81.00%	EAQ4012616.1	12.12
ORF51	-	22,450	22,298	50	hypothetical protein	*Klebsiella* phage ZX4	100%	1.00E-25	94.00%	YP_010054538.1	5.74
ORF52	-	22,817	22,440	125	DUF2591 family protein	*K. pneumoniae*	96%	2.00E-44	63.11%	WP_116961239.1	13.94
ORF53	-	22,948	22,814	44	hypothetical protein	*Klebsiella* phage ZX4	95%	3.00E-17	88.10%	YP_010054539.1	5.15
ORF54	-	23,244	22,945	99	hypothetical protein	*Vibrio* phage pYD38-A	100%	4.00E-60	92.93%	YP_008126235.1	10.99
ORF55	+	23,338	24,045	235	hypothetical protein	*Klebsiella* phage ZX4	100%	7E-171	97.45%	YP_010054540.1	26.55
ORF56	+	24,045	27,338	1097	tail length tape measure protein	*Vibrio* phage pYD38-A	100%	0	96.08%	YP_008126233.1	114.54
ORF57	+	27,338	27,808	156	hypothetical protein	*K. pneumoniae*	100%	3E-112	100.00%	WP_142483272.1	17.99
ORF58	+	27,808	28,278	156	hypothetical protein	*Klebsiella* phage YX3973	100%	1E-109	98.72%	YP_010054357.1	17.93
ORF59	+	28,241	28,699	152	C40 family peptidase	*K. pneumoniae*	100%	2E-107	98.03%	WP_142483270.1	17.40
ORF60	+	28,671	31,127	818	hypothetical protein	*Vibrio* phage pYD38-A	100%	0	99.02%	YP_008126229.1	89.86
ORF61	+	31,166	33,430	754	hypothetical protein	*K. pneumoniae*	95%	0	66.02%	WP_191974704.1	80.67
ORF62	+	33,427	33,672	81	hypothetical protein	*K. pneumoniae*	96%	8.00E-19	47.06%	WP_211775938.1	9.06
ORF63	-	34,175	33,702	157	single-strandedDNA-binding protein	*Vibrio* phage pYD38-A	100%	9.00E-48	52.10%	YP_008126226.1	17.65
ORF64	-	34,844	34,185	219	recombinase	*Klebsiella* phage ZX4	100%	1E-154	98.17%	YP_010054548.1	23.76
ORF65	-	35,824	34,868	318	exonuclease	*Klebsiella* virus KpV2811	100%	0	97.80%	YP_010054422.1	36.00
ORF66	-	36,012	35,824	62	hypothetical protein	*Vibrio* phage pYD38-A	100%	4.00E-37	98.39%	YP_008126223.1	7.13
ORF67	-	36,475	36,059	138	hypothetical protein	*Vibrio* phage pYD38-A	100%	5.00E-95	94.93%	YP_008126222.1	15.74
ORF68	-	38,504	36,462	680	DNA helicase	*Klebsiella* virus KpV2811	100%	0	95.15%	YP_010054420.1	76.95
ORF69	+	38,579	39,472	297	hypothetical protein	*K. pneumoniae*	100%	0	93.94%	WP_142483294.1	33.36
ORF70	-	39,783	39,487	98	hypothetical protein	*Klebsiella* virus KpV2811	100%	7.00E-62	90.82%	YP_010054418.1	11.46
ORF71	-	41,336	39,783	517	hypothetical protein	*Klebsiella* phage ZX4	100%	0	99.81%	YP_010054556.1	57.52
ORF72	-	41,707	41,333	124	hypothetical protein	*Aeromonas* phage pIS4-A	87%	6.00E-59	87.04%	YP_009614674.1	13.87
ORF73	+	42,064	42,285	73	hypothetical protein	*Vibrio* phage pYD38-A	100%	4.00E-40	90.41%	YP_008126215.1	8.05
ORF74	+	42,282	42,467	61	hypothetical protein	*Vibrio* phage pYD38-A	100%	3.00E-36	96.72%	YP_008126214.1	6.76
ORF75	+	42,495	42,761	88	hypothetical protein	*Klebsiella* virus KpV2811	100%	4.00E-55	98.86%	YP_010054492.1	10.03
ORF76	+	42,873	43,220	115	hypothetical protein	*Klebsiella*	99%	9.00E-50	74.56%	WP_004151294.1	12.92
ORF77	+	43,217	43,615	132	hypothetical protein	*K. pneumoniae*	100%	2.00E-94	99.24%	WP_117033037.1	15.28
ORF78	+	43,618	44,235	205	DUF551 domain-containing	*Escherichia coli*	59%	2.00E-60	81.60%	EFM1806727.1	22.84
ORF79	+	44,328	44,486	52	hypothetical protein	*Klebsiella* phage YX3973	100%	9.00E-29	92.31%	YP_010054372.1	6.06
ORF80	+	44,486	44,680	64	hypothetical protein	*Klebsiella* phage YX3973	100%	4.00E-38	95.31%	YP_010054373.1	7.10
ORF81	+	44,784	45,203	139	hypothetical protein	*K. pneumoniae*	99%	5.00E-92	99.28%	WP_142483254.1	15.55
ORF82	+	45,196	45,351	51	hypothetical protein	NO hit	-	-	-	-	5.58
ORF83	+	45,341	46,762	473	large terminase subunit	*Salmonella* phage SeSz-2	100%	0	89.64%	YP_010053627.1	54.99

**Table 3 ijms-23-08040-t003:** Primer information.

	PCR Primers Information
Gene Name	Sequence
** *rpoB* **	**F: Vic3oF:** GTTTTCCCAGTCACGACGTTGTAGGCGAAATGGCWGAGAACCA **R: Vic2oR:** TTGTGAGCGGATAACAATTTCGAGTCTTCGAAGTTGTAACC
** *gapA* **	**F: gapA173oF:** GTTTTCCCAGTCACGACGTTGTATGAAATATGACTCCACTCACGG **R: gapA181oR:** TTGTGAGCGGATAACAATTTCCTTCAGAAGCGGCTTTGATGGCTT
** *mdh* **	**F: mdh130oF:** GTTTTCCCAGTCACGACGTTGTA CCCAACTCGCTTCAGGTTCAG **R: mdh867oR:** TTGTGAGCGGATAACAATTTCCCGTTTTTCCCCAGCAGCAG
** *pgi* **	**F: pgi1FoF:** GTTTTCCCAGTCACGACGTTGTAGAGAAAAACCTGCCTGTACTGCTGGC **R: pgi1RoR:** TTGTGAGCGGATAACAATTTCCGCGCCACGCTTTATAGCGGTTAAT
** *phoE* **	**F: phoE604.1oF:** GTTTTCCCAGTCACGACGTTGTAACCTACCGCAACACCGACTTCTTCGG **R: phoE604.2oR:** TTGTGAGCGGATAACAATTTCTGATCAGAACTGGTAGGTGAT
** *infB* **	**F: infB1FoF:** GTTTTCCCAGTCACGACGTTGTACTCGCTGCTGGACTATATTCG **R: infB1RoR:** TTGTGAGCGGATAACAATTTC CGCTTTCAGCTCAAGAACTTC
** *tonB* **	**F: tonB1FoF:** GTTTTCCCAGTCACGACGTTGTACTTTATACCTCGGTACATCAGGTT **R: tonB2RoR:** TTGTGAGCGGATAACAATTTCATTCGCCGGCTGRGCRGAGAG
**universal sequencing primers**	**F:** GTTTTCCCAGTCACGACGTTGTA **R:** TTGTGAGCGGATAACAATTTC
** *Wzi* **	**F:** GTG CCGCGAGCGCTT TCTATCTTGGTATTCC **R:** GAGAGCCACTGGTTCCAGAATTACCGC
	**qPCR primers information**
**recombinase**	**F:** CCGCTACTCTATTGCGTCTATG **R:** GTTTGCTGCTGAACGAATGAG
**conserved hypothetical protein**	**F:** CCCGGTAATGCGTCAGAATA **R:** TGCCGGATGGTCTGTAATTT
**large terminase subunit**	**F:** GTCGCAATGGTTGAGGTTTATG **R:** TGCCGGATGGTCTGTAATTT

## Data Availability

All data generated for this study are included in the article and Appendix A. All data were analyzed using the GraphPad Prism 8.0.1 and are expressed as means and standard deviation values. Student’s test (*t* test) analysis was used (Figure 2A,C,D). The complete genome sequence of bacteriophage BUCT610 has been deposited in GenBank under the accession number MZ318367.1.

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
