# Peer review of "Characterization and Comparative Genomics Analysis of a New Bacteriophage BUCT610 against Klebsiella pneumoniae and Efficacy Assessment in Galleria mellonella Larvae"

_ijms, 2022, doi:10.3390/ijms23148040_

Round 1

Reviewer 1 Report

Comments to author

Manuscript ID: ijms-1792708

Dear authors,

This is an interesting manuscript and the data presented are relevant to the field of study. However, there are some issues to be addressed. Some suggestions with helpful feedback on the manuscript are below.

Major comments/concerns:

Materials and Methods

A more in-depth description about the source of Klebsiella pneumoniae (“K1119”) was not provided. For example, as a mandatory item, what is the source of Klebsiella pneumoniae, environmental or human?

If the microorganism has been isolated from a human source, an opinion of approval by an ethics committee in research involving humans is mandatory, as well as informed consent, for the collection of microbial samples involving humans. Moreover, if so, the authors must state that “the study was approved by a institutional review board and participants gave their informed consent”, and an institutional review board approval (number) is required, and it must be informed in the Materials and Methods section, regardless of whether such details have been previously published or reported. Otherwise, there is no need only if the source is environmental.   

A better and more detailed methodological description regarding the identification data of Klebsiella pneumoniae should be provided.

Minor comments/concerns:

Title, abstract, and introduction: Change the first cited K. pneumoniae with Klebsiella pneumoniae.

“2.11. Assessment the efficacy of BUCT610 against K. pneumoniae K1119 in vitro”: check this font

Author Response

Dear authors,

This is an interesting manuscript and the data presented are relevant to the field of study. However, there are some issues to be addressed. Some suggestions with helpful feedback on the manuscript are below.

Response: Thanks for your valuable suggestion, we will follow your suggestions to improve the quality of the manuscript.

Major comments/concerns:

Materials and Methods

  1. A more in-depth description about the source of Klebsiella pneumoniae (“K1119”) was not provided. For example, as a mandatory item, what is the source of Klebsiella pneumoniae, environmental or human?

If the microorganism has been isolated from a human source, an opinion of approval by an ethics committee in research involving humans is mandatory, as well as informed consent, for the collection of microbial samples involving humans. Moreover, if so, the authors must state that “the study was approved by a institutional review board and participants gave their informed consent”, and an institutional review board approval (number) is required, and it must be informed in the Materials and Methods section, regardless of whether such details have been previously published or reported. Otherwise, there is no need only if the source is environmental.  

Response: Thanks very much for your reminder and consideration. K. pneumoniae K1119 was isolated from hospital environmental samples and not from human source. And we have described in the manuscript. (Line 62)

  1. A better and more detailed methodological description regarding the identification data of Klebsiella pneumoniae should be provided.

Response: Thanks for your valuable suggestion. We have described in detail the methods for the identification of MLST and capsule types of K. pneumoniae. (Line 94-107; 193-201). And the detailed results are shown in Table 2. The detailed identification of MLST and capsule types of K. pneumoniae is as follows:

“2.6. Multilocus sequence typing (MLST) and capsule type of K. pneumoniae

In order to identify the subtypes of the twenty strains of K. pneumoniae which were used for lytic-range testing. The MLST and capsule type of K. pneumoniae were identi-fied. The determination methods of MLST and capsule type for K. pneumoniae were similar to that described by Yannan Liu et al. and with slight modifications [19]. In brief, seven housekeeping genes (rpoB, gapA, mdh, pgi, phoE, infB, and tonB) and wzi al-lele gene of the twenty K. pneumoniae were subjected to PCR amplification (PCR pro-cedure for MLST identification: step1: 94 ℃ for 2 min; step2: 94 ℃ for 20 s; step 3: 50 ℃ for 30 s; step 4: 72 ℃ for 30 s; step 5: 72 ℃ for 5 min; step 4 to step 2: 35 cycles; PCR procedure for capsule type identification: step1: 94 ℃ for 2 min; step2: 94 ℃ for 30 s; step 3: 55 ℃ for 40 s; step 4: 72 ℃ for 30 s; step 5: 72 ℃ for 5 min; step 4 to step 2: 35 cycles). The amplified products were sent to Beijing Ruibo Xingke Biotechnology Co., Ltd. for bidirectional sequencing. The sequencing results were analyzed by the MLST data-base (https://bigsdb.pasteur.fr/cgi-bin/bigsdb/bigsdb.pl?db=pubmlst_klebsiella_seqdef&l=1) for analysis. Primer sequences are shown in Table 1.

3.2. Multilocus sequence typing and capsule type identification of K. pneumoniae

Seven alleles of rpoB, gapA, mdh, pgi, phoE, infB, and tonB were arranged in that order and analyzed using the multilocus sequence typing scheme that was developed for K. pneumoniae. According to the analysis of MLST Database, the multi-locus sequence type of host bacteria K1119 belong to sequence type 893 (ST893), which belongs to the group of highly virulent K. pneumoniae (hvKp) and has been reported to spread in hospital settings centered on the Intensive Care Unit (ICU) [33,34]. And no matching capsule type of K1119 was identified in the MLST database based on wzi allele sequencing. The multi-locus sequence type and capsule type of other K. pneumoniae were shown in Table 2.”

Minor comments/concerns:

  1. Title, abstract, and introduction: Change the first cited K. pneumoniae with Klebsiella pneumoniae.

Response: Thanks for pointing out our mistakes. We have modified K. pneumoniae to Klebsiella pneumoniae when we first cited Klebsiella pneumoniae in title, abstract, and introduction. (Title, Line 13, Line 22-Line 23, Line 29)

  1. “2.11. Assessment the efficacy of BUCT610 against K. pneumoniae K1119 in vitro”: check this font.

Response: Thanks for pointing out our mistakes. We have modified the font format. (Line 162).

Reviewer 2 Report

Attached

Author Response

Comments on the manuscript entitled “Characterization and comparative genomics analysis of a new bacteriophage BUCT610 against K. pneumoniae and efficacy assessment in Galleria mellonella larvae”

The study has a sound methodology.

The analysis has been described well.

The discussion is clear and balanced.

I have a few minor comments:

Response: Thanks for your valuable suggestion, we will follow your suggestions to improve the quality of the manuscript.

  1. Line 48 Rephrase the sentence to ‘was treated.’

Response: Thanks for pointing out our mistakes. We have modified “is treated” to “was treated” (Line 50).

  1. Line 55 Authors can include a brief description of the KPNE strain K1119 in the introduction.

Response: Thanks very much for your reminder and consideration. We have briefly described K. pneumoniae K1119 in the introduction (Line 56). “K1119” only represents the name of the K. pneumoniae in our laboratory bacterial library. After the identification of MLST and capsule type of K. pneumoniae, the results showed that K1119 belonged to K. pneumoniae ST893 subtype. Therefore, a detailed description of K1119, a strain of K. pneumoniae, was presented in the Results (Line 196-199;). And the specific description of K1119 in the results is as follows:

According to the analysis of MLST Database, the multi-locus sequence type of host bacteria K1119 belong to sequence type 893 (ST893), which belongs to the group of highly virulent K. pneumoniae (hvKp) and has been reported to spread in hospital settings centered on the Intensive Care Unit (ICU) [34,35].

  1. Line156 Standardize the font of ‘efficacy.’

Response: Thanks for pointing out our mistakes. We have modified the font format. (Line 162).

  1. Figure2A Which statistical test was used to determine the P-value

Response: Thank you for your sincere question. The p-value is determined based on the Student's t test (t-test). And we have described in detail the analysis method of the data in 2.13 (Line181-182). The specific description is as follows:

“Student's test (t test) analysis was used in Figure 2A, Figure 2C and Figure 2D.”

  1. Figure 6B Please provide the information on the software used for the analysis of the

survival of G. mellonela

Response: Thanks for your valuable suggestion. The survival of G. mellonela was analyzed using the GraphPad Prism 8.0.1. And we have described in detail the analysis method of the data in 2.13 (Line180 -181). The specific description is as follows:

“All data were analyzed using the GraphPad Prism 8.0.1 and expressed as means and standard deviation values.”
